

# β-asarone induces viability and angiogenesis and suppresses apoptosis of human vascular endothelial cells after ischemic stroke by upregulating vascular endothelial growth factor A

Dazhong Sun[1], Lulu Wu[2], Siyuan Lan[3], Xiangfeng Chi[1] and Zhibing Wu[4]

[1] Department of Acupuncture and Moxibustion Rehabilitation, GuangDong Second Traditional Chinese Medicine Hospital, Guangzhou, China
[2] The First School of Clinical Medicine, Guangzhou University of Chinese Medicine, Guangzhou, China
[3] School of Basic Medicine, Guangzhou University of Chinese Medicine, Guangzhou, China
[4] Department of Neurology, The First Affiliated Hospital of Guangzhou University of Chinese Medicine, Guangzhou, China

Corresponding author
Zhibing Wu,
drwuppsub@outlook.com

## ABSTRACT

Ischemic stroke (IS) is a disease with a high mortality and disability rate worldwide, and its incidence is increasing per year. Angiogenesis after IS improves blood supply to ischemic areas, accelerating neurological recovery. β-asarone has been reported to exhibit a significant protective effect against hypoxia injury. The ability of β-asarone to improve IS injury by inducing angiogenesis has not been distinctly clarified. The experimental rats were induced with middle cerebral artery occlusion (MCAO), and oxygen-glucose deprivation (OGD) model cells were constructed using human microvascular endothelial cell line (HMEC-1) cells. Cerebral infarction and pathological damage were first determined *via* triphenyl tetrazolium chloride (TTC) and hematoxylin and eosin (H&E) staining. Then, cell viability, apoptosis, and angiogenesis were assessed by utilizing cell counting kit-8 (CCK-8), flow cytometry, spheroid-based angiogenesis, and tube formation assays in OGD HMEC-1 cells. Besides, angiogenesis and other related proteins were identified with western blot. The study confirms that β-asarone, like nimodipine, can ameliorate cerebral infarction and pathological damage. β-asarone can also upregulate vascular endothelial growth factor A (VEGFA) and endothelial nitric oxide synthase (eNOS) and induce phosphorylation of p38. Besides, the study proves that β-asarone can protect against IS injury by increasing the expression of VEGFA. *In vitro* experiments affirmed that β-asarone can induce viability and suppress apoptosis in OGD-mediated HMEC-1 cells and promote angiogenesis of OGD HMEC-1 cells by upregulating VEGFA. This establishes the potential for β-asarone to be a latent drug for IS therapy.

## INTRODUCTION

Stroke is an acute disorder of cerebral blood circulation with high morbidity, recurrence, disability, and mortality (*Wilson & Ashcraft, 2023*). It is broadly categorized as ischemic stroke (IS) and hemorrhagic stroke, where IS accounts for 60–80% of the total stroke incidence (*Xu et al., 2022*). Globally, about 795,000 people experience a new or recurrent stroke each year, and 87% of strokes are ischemic (*Barthels & Das, 2020*). Currently, the only thrombolytic therapy for IS approved by the Food and Drug Administration (FDA) is the tissue plasminogen activator (tPA) (*Kadir & Bayraktutan, 2020*). tPA can improve cerebral blood flow and alleviate brain tissue damage, but it can increase the risk of hemorrhagic transformation and cerebral edema (*Shah, Paul & Yadav, 2024*). Only about 8% of patients with IS are eligible for tPA (*Shademan et al., 2023*). A previous study confirmed that cerebral apoplexy can induce reactive angiogenesis in ischemic areas (*Hatakeyama, Ninomiya & Kanazawa, 2020*). The formation of new vessel networks helps ameliorate blood perfusion and accelerate the functional recovery of damaged nerves (*Tsivgoulis et al., 2023*). Therefore, drug administration or gene intervention to effectively induce therapeutic angiogenesis can fundamentally resolve the dilemma of IS therapy.

*Acorus tatarinowii Schott* has been reported to exhibit multiple beneficial properties, such as promoting the restoration of special sense or consciousness, removing dampness to restore stomach homeostasis, awakening spirit, enhancing intelligence, and so forth (*Lee, Kao & Cheng, 2020*; *Wang et al., 2020*). The main active components of *Acorus tatarinowii Schott* are α-asarone and β-asarone, both of which have spasmolytic, anticonvulsant, and synergistic effects when combined with pentobarbital sodium (*Pu et al., 2023*). In particular, β-asarone has multiple pharmacological uses (*Tao et al., 2020*). Clinically, β-asarone is utilized to treat fever, delirium, phlegm, forgetfulness, dementia, deafness, tinnitus, heartburn, traumatic injury, and so on (*Hei et al., 2020*; *Saki et al., 2020*). The cascade reactions induced by cerebral ischemia include the release of excitatory amino acids, apoptosis, intracellular calcium influx overload, inflammation, activation of calcium-dependent enzymes, nitric oxide free radical production, and so on (*She et al., 2023*; *Shin et al., 2020*; *Tian et al., 2023*). β-asarone has been confirmed to have antioxidant and anti-inflammatory properties (*Shi et al., 2021*). It is one of the most frequently utilized compounds in treating nervous system disorders because it can successfully cross the blood-brain barrier (BBB) and enter the brain tissue (*Pan et al., 2021*). It has also been established that β-asarone can prevent neuronal apoptosis and enhance cognitive function in mice (*Balakrishnan et al., 2022*). Additionally, β-asarone can also improve brain neural diseases, Alzheimer's disease, epilepsy, depression, and so on (*Wang et al., 2021*). Recent research has substantiated that β-asarone can alleviate cerebral infarction in rats with middle cerebral artery occlusion (MCAO) rats (*Pan et al., 2021*). However, the role of β-asarone in ameliorating IS has not yet been established.

Recent studies proved angiogenesis to be a key protective mechanism that promotes neuroregeneration and functional recovery during IS pathology (*Fang, Wang & Miao, 2023*; *Shi et al., 2023*). Following the rupture of the BBB in IS, reactive oxygen species trigger morphological changes in astrocytes to form reactive astrocytes, which can alter the

extracellular matrix (ECM), leading to complete remodeling of the ECM and the formation of ECM bundles (*Williamson et al., 2021*). The endothelial cells then migrate through the ECM bundles to establish new capillary sprouts (*Arbaizar-Rovirosa et al., 2023*). Therefore, induction of endothelial cell angiogenesis can effectively improve vascular remodeling and neurovascular function recovery after IS. A recent study has validated that α-asarone can enhance tubular formation and induce angiogenesis (*Balakrishnan et al., 2022*). Besides, it can also accelerate functional recovery in rats with spinal cord injury (*Jo et al., 2018*). However, it remains unclear whether β-asarone can induce angiogenesis in IS.

This study utilized MCAO model rats and oxygen-glucose deprivation (OGD) model HMEC-1 cells to thoroughly examine the medicinal value of β-asarone. The influence of β-asarone on cerebral infarction and pathological damage in MCAO model rats was confirmed. Meanwhile, the impacts of β-asarone on the viability, apoptosis, and angiogenesis of OGD-mediated HMEC-1 cells were also discovered. More importantly, this study proved whether the mechanism of β-asarone involved in angiogenesis is realized through the VEGF pathway. Therefore, this study aims to provide a basis for IS therapy by exploring the structure and function of β-asarone.

## MATERIALS AND METHODS

### Animals

Healthy adult Sprague-Dawley (SD) rats (male, 6–8 weeks, weighing 220–260 g) without genetic modifications were provided by the first affiliated hospital of the Guangzhou University of Chinese Medicine Animal Experiment Center. All rats were kept in the first affiliated hospital in separate cages maintained at 20–25 °C with 40–70% humidity, 12 h of light a day, solid diets, and clean drinking water. All animal experiments were approved by the Ethics Committee of the first affiliated hospital of Guangzhou University of Chinese Medicine (No. GZTCMF1-2022038). The animal experiments were conducted as per the established guidelines for the care and use of laboratory animals. A single animal served as the experimental unit in this study. All assays were carried out in the core lab.

### Establishment of the MCAO model

The MCAO model was created by referring to the methodology of a previous study (*Yu et al., 2020a*). Prior to the surgery, the SD rats were fasted for 12 h and were free to exercise. They were then weighed, anchored to a small operating table, and anesthetized with an intraperitoneal injection of 1% pentobarbital sodium (Sigma, St Louis, MO, USA). Further, the neck hair was removed, and the neck region was disinfected with 1% iodine. An incision was made in the middle of the neck, and the subcutaneous tissue of the neck was separated to expose the deep neck structure. The external carotid artery (ECA), common carotid artery (CCA), and internal carotid artery (ICA) were exposed under a surgical microscope. The CCA and ECA were then ligated, and the ICA was temporarily clipped with an arterial clamp. Subsequently, an inverted V-shaped incision was made on the proximal wall of the ECA. A 4-0 nylon thread was inserted at the beginning of the middle cerebral artery (MCA) to block the blood flow of the MCA. The rectal temperature was maintained between 36.5 °C to 37 °C during the whole process. After 2 h of embolization,

the threaded plug was removed, the wound was sutured, and the rats were put back into the cage for free feeding with water. Additionally, the Zea Longa score of rats was analyzed after they woke up. The rats with neurological dysfunction scores of 1–4 were classified as effective models, and the rats with scores of 0 and 5 were classified as failure models and were eliminated. Two rats were excluded during MCAO modeling. After exclusion, the rats were randomly replenished within each group to ensure each group had five rats (the number was unchanged).

## β-asarone administration in rats

Based on the correlational research, β-asarone was obtained from *Acorus tatarinowii Schott*, which also was analyzed using gas chromatography-mass spectrometry (GC-MS), nuclear magnetic resonance, and infrared spectroscopy. The purity of β-asarone was determined to be 99.55%. MCAO model rats were administered 30 mg/kg of β-asarone twice a day for 4 weeks (*Yang et al., 2017*).

## Nimodipine administration in rats

The rats were administered 8 mg/kg nimodipine (Bayer, Leverkusen, Germany) by gavage 30 min before MCAO (*Sun et al., 2023*; *Zhang et al., 2022*). This study utilized Nimodipine as a positive control drug.

## Intracerebral stereotactic injection of adenovirus

After weighing the SD rats, they were intraperitoneally anesthetized with 1% pentobarbital (0.45 mL/100 g) and fixed on a DY-II animal brain stereograph. The scalps of the rats were disinfected with 1% iodine and dissected in the middle of the head. The anterior fontanelle was exposed, and the skull surface was drilled to the right of the midline. Then, an adenovirus suspension (10 μL, short hairpin negative control (sh-NC), and sh-vascular endothelial growth factor A (VEGFA)) was slowly injected into the striatum (0.8 mm behind bregma, 3 mm beside the sagittal suture, and 4.5 mm subdurally) with a microinjector for 10 min, and the needle was left for 5 min after injection (*Du et al., 2022*). Postoperatively, the bone window was closed, and the surgical incision was sutured. After emergence from anesthesia, the rats were returned to the original cage for further feeding.

## Animal grouping

All rats were randomly assigned to each group. A computer program was utilized to generate random numbers and divide the rats into different groups according to the generated numbers. This made it unclear whether each rat would be in the experimental or control group. The SD rats were randomly divided into the sham group, MCAO, MCAO + nimodipine (Nim), and MCAO + β-asarone groups. Besides, they were randomly divided into the MCAO + β-asarone, MCAO + β-asarone + sh-NC, and MCAO + β-asarone + sh-VEGFA groups. A total of 40 rats were divided into seven groups, with each group containing five animals. Five rats were reserved to replenish those that failed to model. After successful modeling, the relevant experiments were performed using the brain tissues (10 mg). The following analyses were conducted: modified neurological severity score (mNSS) determination, triphenyl tetrazolium chloride (TTC) staining (cerebral infarction

area), hematoxylin and eosin (H&E) staining (pathological structure of brain tissue), quantitative polymerase chain reaction (qPCR) (to analyze VEGFA expression in brain tissue), western blot (to analyze VEGFA, p38, p-p38, endothelial nitric oxide synthase (eNOS) and glyceraldehyde-3-phosphate dehydrogenase (GAPDH) expression in brain tissue), immunofluorescence assay (to analyze CD31 and Ki67 expression in brain tissue).

## Treatment of rats

All rats were euthanized using a $CO_2$ euthanasia device (PVC, Wonderful Oasis Biotechnology, China). $CO_2$ was injected into the euthanasia chamber at a rate that replaced 10% to 30% of the chamber's volume per minute. It was ensured that the rats were immobile, breathing ceased, and pupils were dilated. Following this, $CO_2$ input was discontinued, and the rats were observed for another 5 min to ascertain their death. At the experimental endpoint, there were no surviving animals. The brain tissue was removed by craniotomy and frozen at −20 °C for 10 min in a refrigerator.

## TTC staining

The brains were removed by decapitation and cut along the coronal plane into 2 mm thick sections. The samples were then fixed with 2% TTC solution (Sigma, St Louis, MO, USA) at 37 °C for 30 min away from light. After fixing, the brain sections were observed and photographed, and the infarct area was confirmed with Image-Pro Plus. The percentage of cerebral infarction area was calculated as follows:

Percentage of cerebral infarction area = Total cerebral infarction area/Total cerebral slice area × 100%.

## mNSS neurological function score

mNSS was adopted to evaluate the neurological loss and recovery of ischemic rats in a single-blind study. The mNSS neurological function scores ranged between 0 to 18 (normal 0; maximal deficit score 18). mNSS considers motion, sensation, reflexes, and balance. The higher the score, the more severe the injury.

## Hematoxylin and eosin staining

The brain tissues were fixed with 10% formalin (Thermo Fisher Scientific, Waltham, MA, USA), dehydrated with gradient alcohol (Sigma, St Louis, MO, USA), cleared with xylene (Sigma-Aldrich Co, St. Louis, MO, USA), embedded in paraffin, and sectioned. Then, the brain sections were dried in an incubator maintained at 45 °C, dewaxed in xylene, and dehydrated with alcohol from high to low concentrations. After soaking in distilled water, the brain sections were stained with hematoxylin (Cat. no. H3136; Sigma-Aldrich, St. Louis, MO, USA) for 10 min, differentiated with 1% hydrochloric acid alcohol for a moment, rinsed with distilled water for 1 h, and stained with eosin (Cat. no. 6766007; Thermo Fisher Scientific, Waltham, MA, USA) for 3 min. After dyeing, the sections were dehydrated with pure alcohol, cleared with xylene, and sealed with neutral gum. Finally, the pathological tissue was observed under a light microscope.

## Immunofluorescence double-staining

The rat brain tissue was completely removed after heart perfusion and was subsequently fixed by 4% paraformaldehyde (Sigma Chemical Co., St Louis, MO, USA), dehydrated by 20% and 30% sucrose, and embedded. The brain tissue was then cut into 4 μm thick sections with a frozen slicer. After cleaning with phosphate-buffered saline (PBS), the brain sections were treated with 1% bovine serum albumin (BSA) (Sigma-Aldrich, St. Louis, MO, USA), sealed for 1 h, and exposed to appropriate CD31 (Santa Cruz, Dallas, TX, USA, 1:50) and Ki67 antibodies (Cell Signaling Technology, Danvers, MA, USA, 1:200) at 4 °C overnight. The next day, the brain sections were treated with AlexaFluor 594 IgG (1:200) and AlexaFluor488 IgG (1:200) and incubated away from light for 1 h. After washing, the sections were stained with an appropriate amount of 4′,6-diamidino-2-phenylindole (DAPI) (1:30), stored away from light for 10 min, and sealed with 50% glycerol. The results were photographed with a fluorescence microscope (Nikon, Tokyo, Japan). The Ki67 and CD31 positive cells were counted through observation in each field.

## Cell culture

The human vascular endothelial cell line (HMEC-1) was acquired from the American Type Culture Collection (ATCC, Manassas, VA, USA). They were grown in an MCDB 131 medium containing 10% fetal bovine serum (FBS) (Sigma, St Louis, MO, USA), 10 ng/mL epidermal growth factor (Gibco, Grand Island, NY, USA), 2 mM glutamine (Sigma, Welwyn Garden City, UK), and 1 μg/mL hydrocortisone (Sigma, Welwyn Garden City, UK) at 37 °C with 5% $CO_2$.

## Establishment of the OGD cell model

HMEC-1 cells (about 85% fusion degree) were collected, centrifuged, and cleaned with 2 mL PBS (Gibco, Grand Island, NY, USA). Then the cells ($1 \times 10^5$ cells/well) were evenly spread into 12-well plates and cultured for 24 h. Subsequently, the cells were added to 1 mL of sugar-free medium and placed in a hypoxic chamber with 95% $N_2$, 1% $O_2$, and 4% $CO_2$ for 3 h. Then the sugar-free medium was replaced by a complete culture with high glucose, and the cells were cultured at 37 °C with 5% $CO_2$ for 24 h of reoxygenation. Cells in the normal group were routinely cultured until the end of reoxygenation in other groups.

## Cell treatment

OGD-treated HMEC-1 cells were treated with nimodipine (10 μmol/L) (*Zech et al., 2020*) or β-asarone (20, 30, and 45 μg/mL), respectively (*Mo et al., 2012*). Negative control (si-NC) and VEGFA small interfering ribonucleic acid (siRNA) (si-VEGFA) were sourced from GenePharma (Shanghai, China). The sequence of si-NC used was 5′-AATTCTCCGAACGTGTCACGT-3′; the sequence of si-VEGFA used was 5′-AUGUGAAUGCAGACCAAAGAA-3′. OGD-treated HMEC-1 cells ($1 \times 10^5$ cells/well) in a 6-well plate and incubated overnight. 5 μL of siRNA was mixed with 250 μL serum-free medium for 5 min; 5 μL Lipofectamine™ 3000 (Invitrogen, Waltham, MA, USA) was mixed with 250 μL serum-free medium for 5 min; the above mixtures were further mixed for 20 min. The complex mixture was added to six-well plates and incubated for 6 h. After

replacing the complex mixture with a complete medium, the culture was continued for 48 h.

## RNA extraction and real-time quantitative PCR

Brain tissues (2 mg) in each group were ground on ice and treated with 500 μL TRIzol (Invitrogen Life Technologies, Carlsbad, CA, USA) to extract the RNA. The genome was digested with RNase-free DNase I (Promega, Beijing, China). The purity (A260/A280 ratio) of RNA was between 1.8 and 2.0, as determined using an ultraviolet/visible (UV/Vis) spectrophotometer (BioPhotometer Plus; Eppendorf, Hamburg, Germany). A total of 40 μg of RNA was extracted, which was stored at −80 °C in a refrigerator. Agarose gel electrophoresis was performed to analyze RNA integrity, which produced intact 28S and 18S ribosomal RNA (rRNA) bands. The Agilent 2200 RNA assay (Agilent Technologies, Inc., Santa Clara, CA, USA) was employed to detect the RNA integrity, and an RNA integrity number ≥7 was considered ideal. A UV/Vis spectrophotometer (BioPhotometer Plus; Eppendorf, Hamburg, Germany) was utilized to analyze RNA concentration. Then, reverse transcription was conducted using the RevertAid First Strand cDNA Synthesis Kit (Cat. no. K1622; Thermo Scientific, Waltham, MA, USA) with 1.0 μg of RNA. The 20 μl mixture of the reverse transcription reaction included 1 μL RNA, 5 μL random primer, 4 μL 5× reaction buffer, 2 μL deoxynucleoside triphosphates (dNTPs) (10 mM), 1 μL RiboLock RNase Inhibitor, 1 μL RevertAid Reverse Transcriptase, and 2 μL nuclease-free water. The reverse transcription reaction conditions were 25 °C for 5 min, 42 °C for 60 min, and 70 °C for 10 min. The complementary deoxyribonucleic acid (cDNA) was stored at −20 °C in a refrigerator. The amplification reaction was conducted with SYBR green PCR Master Mix (Applied Biosystems, Warrington, UK) and an ABI 7500 Real-Time PCR system (Applied Biosystems). The 10 μL mixture of the real-time PCR reaction included 5.0 μL SYBR® Premix Ex Taq$^{TM}$ II (2×), 0.4 μL PCR forward primer (10 μM), 0.4 μL PCR reverse primer (10 μM), 0.2 μL 6-carboxyl-X-Rhodamine (ROX) reference dye (50×), 1.0 μL cDNA, and 3.0 μL of double distilled water (ddH$_2$O). The real-time PCR reaction setup (manual) was as follows: pre-denaturation at 95 °C for 30 s (one cycle), denaturation at 95 °C for 5 s, and annealing at 60 °C for 34 s (40 cycles). The amplicon length was 100 base pairs (bp). The sequences of primers were: VEGFA (NM_001110333.2) forward: 5′-AGAAAGCCCATGAAGTGGTGA-3′ and reverse: 5′-TCTCATCGGGGTACTCCTGG-3′; glyceraldehyde 3-phosphate dehydrogenase (GAPDH) forward: 5′-CCGCATCTTCTTGTGCAGTG-3′ and reverse: 5′-CGATACGGCCAAATCCGTTC-3′. Melting curve analysis was conducted at a temperature range of 60–95 °C. The slope fluctuated between −3.59 and 3.1, with an $R^2 \geq 0.9$ indicating reliable results. The relative expression of VEGFA was assessed using the $2^{-\Delta\Delta Ct}$ method. GAPDH was the internal reference for VEGFA. All experiments were independently repeated thrice.

## Western blot

The proteins were isolated from the ground brain tissues and the processed HMEC-1 cells using a radio-immunoprecipitation assay (RIPA) buffer (Beyotime, Shanghai, China). After quantification utilizing the bicinchoninic acid (BCA) kit (Fude Biological

Technology Co., Ltd., Hangzhou, China), the proteins (40 µg) were subjected to electrophoresis (10% sodium dodecyl-sulfate polyacrylamide gel electrophoresis (SDS-PAGE), Elabscience, Wuhan, China) and electrically transferred to a polyvinylidene difluoride (PVDF) membrane (Millipore, Burlington, MA, USA). Then, the membranes were sealed (5% skim milk) for 1 h, incubated with primary antibodies, and exposed to secondary antibodies (1:3,000; ab288151, Abcam, Cambridge, MA, USA) for 1 h. After staining with the enhanced chemiluminescence solution (ECL; Cat. no. RPN3001; Amersham Biosciences, Piscataway, NJ, USA), western blotting was performed. The primary antibodies contained anti-VEGFA (1:1,000, ab155944; Abcam, Cambridge, MA, USA), anti-T-p38 (1:2,000, ab170099; Abcam, Cambridge, MA, USA), anti-P-p38 (1:2,000, ab4822; Abcam, Cambridge, MA, USA), anti-eNOS (1:2,000, ab76198; Abcam, Cambridge, MA, USA), and anti-GAPDH (1:2,000, ab9485; Abcam, Cambridge, MA, USA).

### Cell counting kit-8

The processed HMEC-1 cells ($1 \times 10^4$ cells/well) were seeded into 96-well plates and incubated at 37 °C. The cell counting kit-8 (CCK-8) (Solarbio, Beijing, China) was utilized to measure cell viability at 48 h in a ratio of 10 µL of CCK-8 to 100 µL of the mixture. After incubation at 37 °C for 2 h, the optical density (OD) was measured at 450 nm on a microplate meter.

### Spheroid-based angiogenesis assay

The HMEC-1 cells were re-suspended using the medium with 0.20% (weight/volume) carboxymethyl cellulose (Sigma-Aldrich, St. Louis, MO, USA) and then uniformly inoculated into a 96-well plate with a non-adherent round bottom. This facilitated HMEC-1 cells to form a pellet of a fixed diameter and number of cells. After cell pellets were formed overnight, they were coated with Matrigel™ (BD, Franklin Lakes, NJ, USA) without the growth factor, transferred to a pre-cooled, flat-bottom 96-well plate, and maintained at 37 °C for 30 min. The cells were then routinely cultured for 24 h and photographed.

### Tube formation assay

The precooled Matrigel™ (100 µL, BD, Franklin Lakes, NJ, USA) was gently added to a 96-well plate and stored at 37 °C for 30 min. The processed HMEC-1 cells were collected and inoculated into 96-well plates at a density of $1 \times 10^4$ cells/well at 37 °C for 16 h. The tubule formation of cells was observed under a light microscope. Capillary lumen-like structures were also analyzed by WimTube software (Ibidi, Gräfelfing, Germany).

### Statistical analysis

All experiments were independently repeated thrice, and the data are displayed as mean ± standard deviation (SD). The data evaluated met the statistical assumptions. The Statistical Package for Social Sciences (SPSS) 21.0 software (SPSS Inc, Chicago, IL, USA) was utilized to conduct a one-way analysis of variance (ANOVA) with Tukey's *post-hoc* test (α = 0.05). *P* values < 0.05 were considered statistically significant.

## RESULTS

### Similar to nimodipine, β-asarone significantly reduced infarct area and alleviated pathological injury in MCAO model rats

To determine the function of β-asarone in IS, this study first constructed MCAO model rats. Subsequently, the rats were administered β-asarone by gavage for 4 weeks (Fig. 1A). TTC staining results revealed that the cerebral infarct area was signally aggrandized in the MCAO model group *versus* that in the sham group, and nimodipine or β-asarone treatment notably reduced the cerebral infarct area in MCAO model rats (Figs. 1B and 1C). Simultaneously, it was discovered that the MCAO model rats displayed contralateral limb paralysis and other neurological defects. The grading concluded that relative to the sham group, mNSS was evidently increased in the MCAO model group. However, this elevation in mNSS can be substantially reduced by nimodipine or β-asarone in MCAO model rats, indicating that β-asarone, like nimodipine, can ameliorate neurobehavioral function in MCAO model rats (Fig. 1D). Besides, the results of H&E staining demonstrated that in the sham group, the morphology and structure of nerve cells were normal, with prominent nucleoli and no inflammatory cell infiltration. In the MCAO model group, the structure of the ischemic area in brain tissue was loose, vacuolated, and exhibited interstitial edema. In nimodipine and β-asarone treatment groups, the pathological damage of brain tissue was alleviated as expected (Fig. 1E). Overall, these data verify that β-asarone, like nimodipine, can ameliorate infarction and pathological injury in MCAO model rats.

### β-asarone markedly regulated VEGFA, p38 MAPK, and eNOS pathways in MCAO model rats

The study further explored the possible pathways that β-asarone might regulate in MCAO model rats. RT-qPCR results demonstrated that VEGFA was dramatically downregulated in the MCAO group as compared to that in the sham group, which could be partially restored by nimodipine or, in particular, β-asarone (Fig. 2A). Meanwhile, western blot data indicated that VEGFA and eNOS were significantly downregulated, and p-p38 was markedly upregulated in the MCAO group *versus* that in the sham group, which could be markedly restored using nimodipine or β-asarone (Figs. 2B and 2C). Besides, this study found that nimodipine or β-asarone can evidently increase CD31 and Ki-67 positive cells in MCAO model rats (Figs. 2D and 2E). Thus, this highlights how the protective effect of β-asarone on brain injury in rats may be realized through VEGFA, p38, and eNOS.

### VEGFA silencing dramatically increased infarct area and pathological injury in MCAO model rats mediated by β-asarone

Because β-asarone upregulated VEGFA expression, the study further investigated whether VEGFA exhibits protective effects along with β-asarone on IS. VEGFA was silenced in the brain of MCAO model rats by intracerebral stereotactic injection of lentivirus (Fig. 3A). As depicted in Figs. 3B and 3C, VEGFA silencing markedly increased the cerebral infarct area of MCAO model rats, which has been reduced by β-asarone. The decreased mNSS caused by VEGFA silencing in MCAO model rats can be signally increased by β-asarone

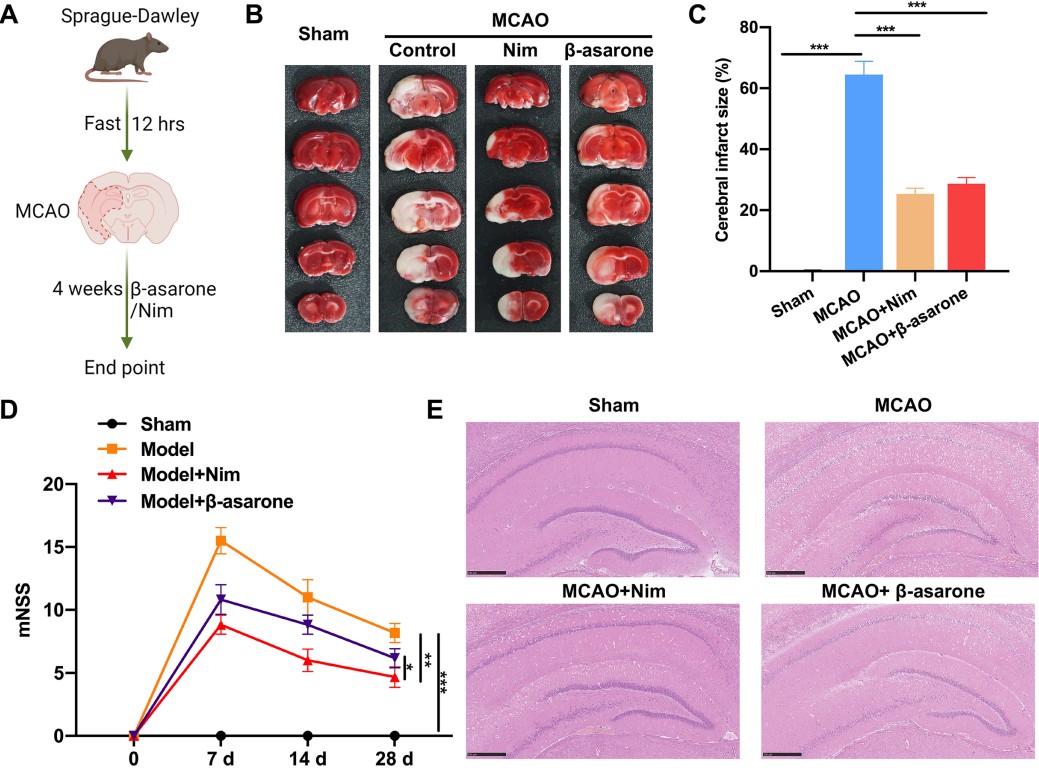

**Figure 1 Similar to nimodipine, β-asarone significantly reduced infarct area and alleviated pathological injury in middle cerebral artery occlusion (MCAO) model rats.** (A) After fasting for 12 h, the MCAO model was induced in rats, and they were nimodipine or β-asarone was administered for 4 weeks. Image credit: the BioRender at https://app.biorender.com/. (B) Triphenyl tetrazolium chloride (TTC) staining demonstrated the change of cerebral infarct area in the brain tissues of rats. (C) The cerebral infarct size was measured using the results of TTC staining. (D) The behavior changes in rats were evaluated by mNSS. (E) Hematoxylin and eosin (H&E) staining was performed to assess the pathological structure in the brain tissues of rats. Magnification: 200×, scale bar = 50 μm. $^{*}P < 0.05$, $^{**}P < 0.01$, $^{***}P < 0.001$.

(Fig. 3D). Subsequently, H&E staining demonstrated that the reduction of pathological damage aided by β-asarone could be markedly reversed by VEGFA silencing in MCAO model rats (Fig. 3E). As a whole, these results confirm that upregulation of VEGFA is essential to aid β-asarone in alleviating IS injury.

## VEGFA silencing evidently induced phosphorylation of p38 and downregulated eNOS in MCAO model rats mediated by β-asarone

This study further confirms that silencing VEGFA dramatically downregulated eNOS in MCAO model rats (Figs. 4A and 4B). Next, western blot results confirmed that VEGFA silencing significantly upregulated p-p38 and downregulated eNOS in MCAO model rats (Figs. 4B and 4C). Besides, IF results demonstrated that silencing VEGFA significantly decreased CD31 and Ki-67 positive cells mediated by β-asarone in MCAO model rats (Figs. 4D and 4E). In short, this study proves that VEGFA silencing can also reverse the regulation of p-p38 and eNOS expressions mediated by β-asarone in MCAO model rats.

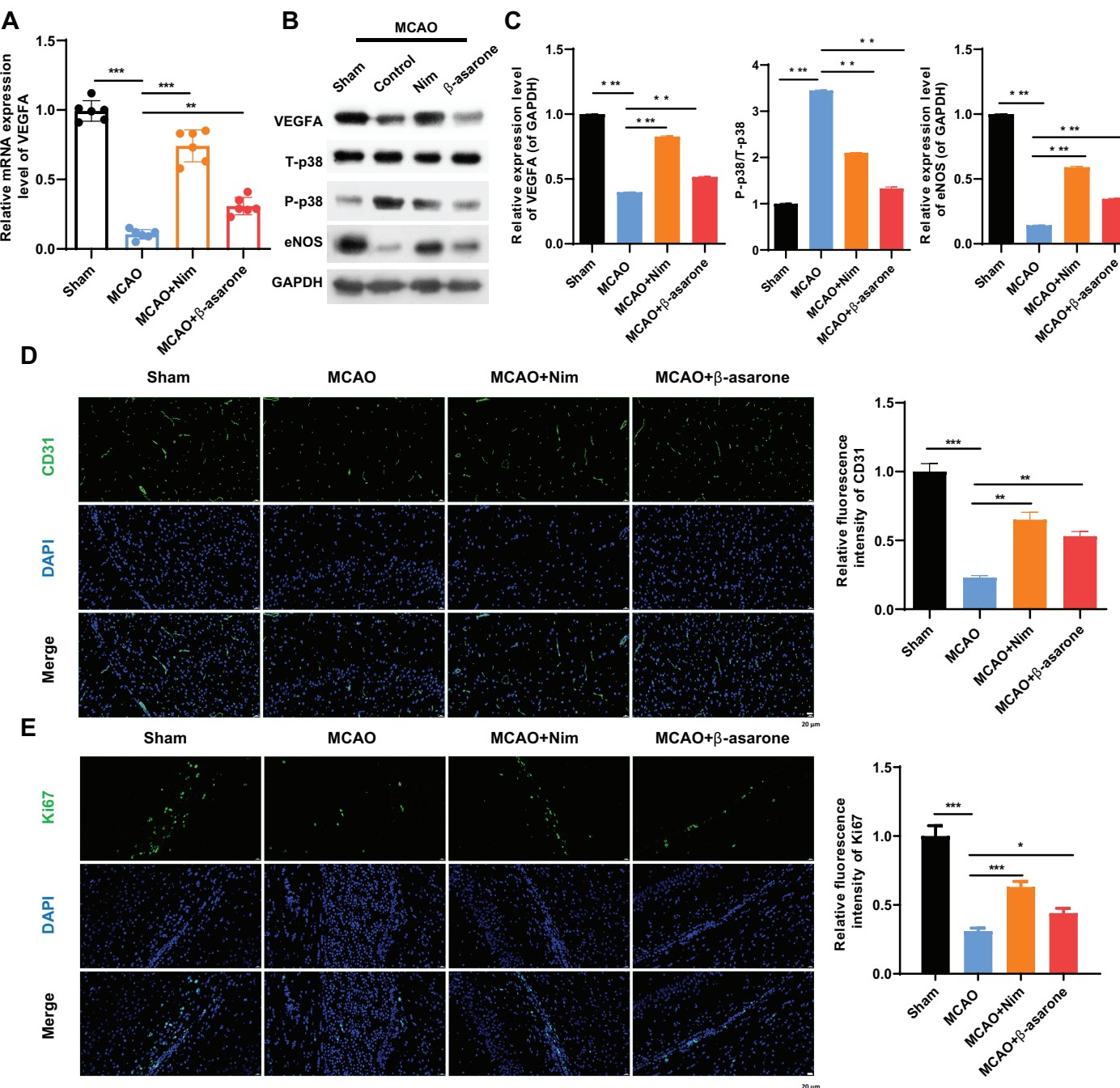

**Figure 2 β-asarone notably regulated vascular endothelial growth factor A (VEGFA), p38 MAPK, and endothelial nitric oxide synthase (eNOS) pathways in MCAO model rats.** (A) The change in VEGFA expression was monitored by RNA extraction and real-time quantitative PCR (RT-qPCR) in the brain tissues of the MCAO model rats after administering nimodipine or β-asarone. (B) Western blot helped assess the impacts of nimodipine or β-asarone on the VEGFA, T-p38, P-p38, and eNOS expressions. (C) Relative quantification of proteins was calculated using the data obtained from the western blot. (D) The variations in expressions CD31 and Ki67 were tested through IF double-staining. Magnification: 400×, scale bar = 25 μm. (E) CD31 and Ki67 positive cells were quantitatively analyzed. Ns, No statistical significance; *$P < 0.05$, **$P < 0.01$, ***$P < 0.001$.

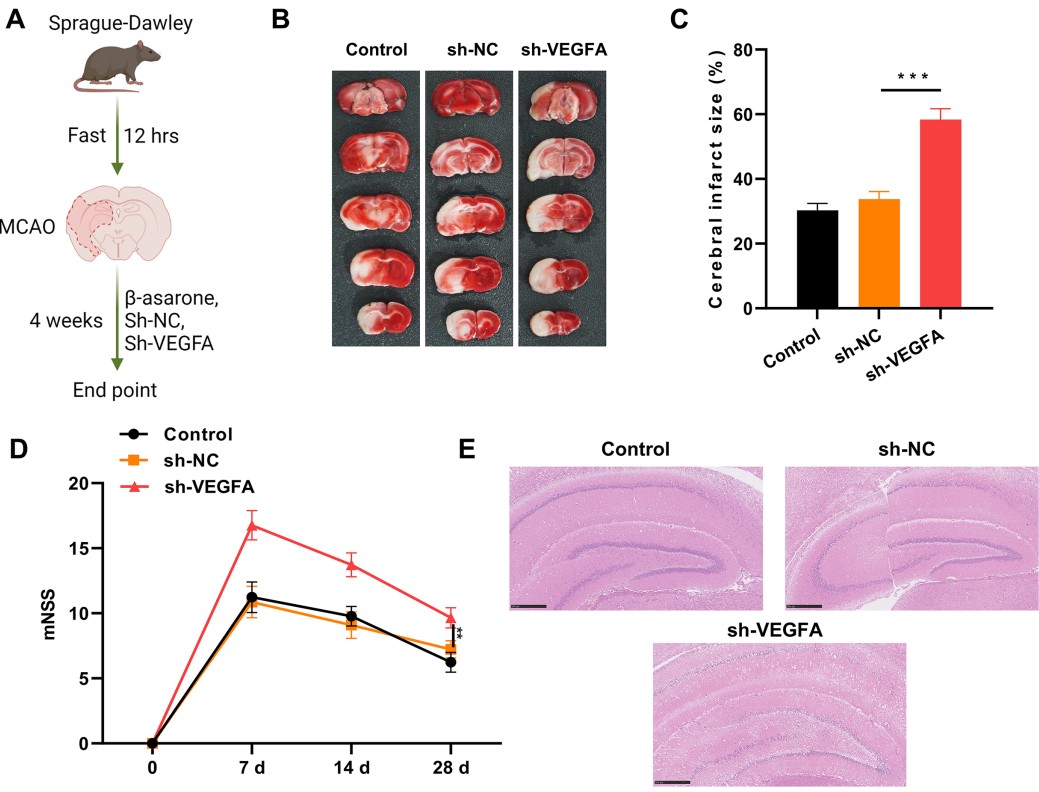

**Figure 3 VEGFA silencing dramatically increased infarct area and pathological injury in MCAO model rats mediated by β-asarone.** (A) VEGFA was silenced in MCAO model rats after β-asarone administration for 4 weeks through the intracerebral stereotactic injection of adenovirus. Image credit: the BioRender at https://app.biorender.com/. (B) The change in cerebral infarct area was ascertained through TTC staining in MCAO model rats. (C) The cerebral infarct size was then measured. (D) mNSS was adopted to assess the change in rat behavior. (E) H&E staining displayed changes in the pathological structure. Magnification: 200×, scale bar = 50 μm. $^{**}P < 0.01$, $^{***}P < 0.001$. Control and MCAO model rats were administered β-asarone.

## β-asarone evidently accelerated cell viability and angiogenesis and suppressed apoptosis in HMEC-1 cells under OGD

To further clarify the impact of β-asarone on IS *in vitro*, an OGD cell model was constructed using HMEC-1 cells. As represented in Fig. 5A, OGD caused a remarkable reduction in cell viability, which could be evidently restored by nimodipine or β-asarone administration in HMEC-1 cells. OGD also resulted in an outstanding elevation in the apoptosis of HMEC-1 cells, which could also be signally weakened by nimodipine or β-asarone, especially 30 μg/mL of β-asarone; higher concentrations of β-asarone can cause certain damage to HMEC-1 cells under OGD (Figs. 5B and 5C). Therefore, 30 μg/mL of β-asarone was utilized to treat HMEC-1 cells in subsequent experiments, and HMEC-1 cells under the OGD condition were processed with nimodipine (10 μmol/L) or β-asarone (30 μg/mL) (Fig. 5D). As VEGFA is associated with angiogenesis, this study further verified whether β-asarone can affect cell angiogenesis in OGD-induced HMEC-1 cells. Firstly, the study uncovered that OGD markedly downregulated VEGFA and eNOS and strengthened p38 phosphorylation in HMEC-1 cells, which could be reversed with

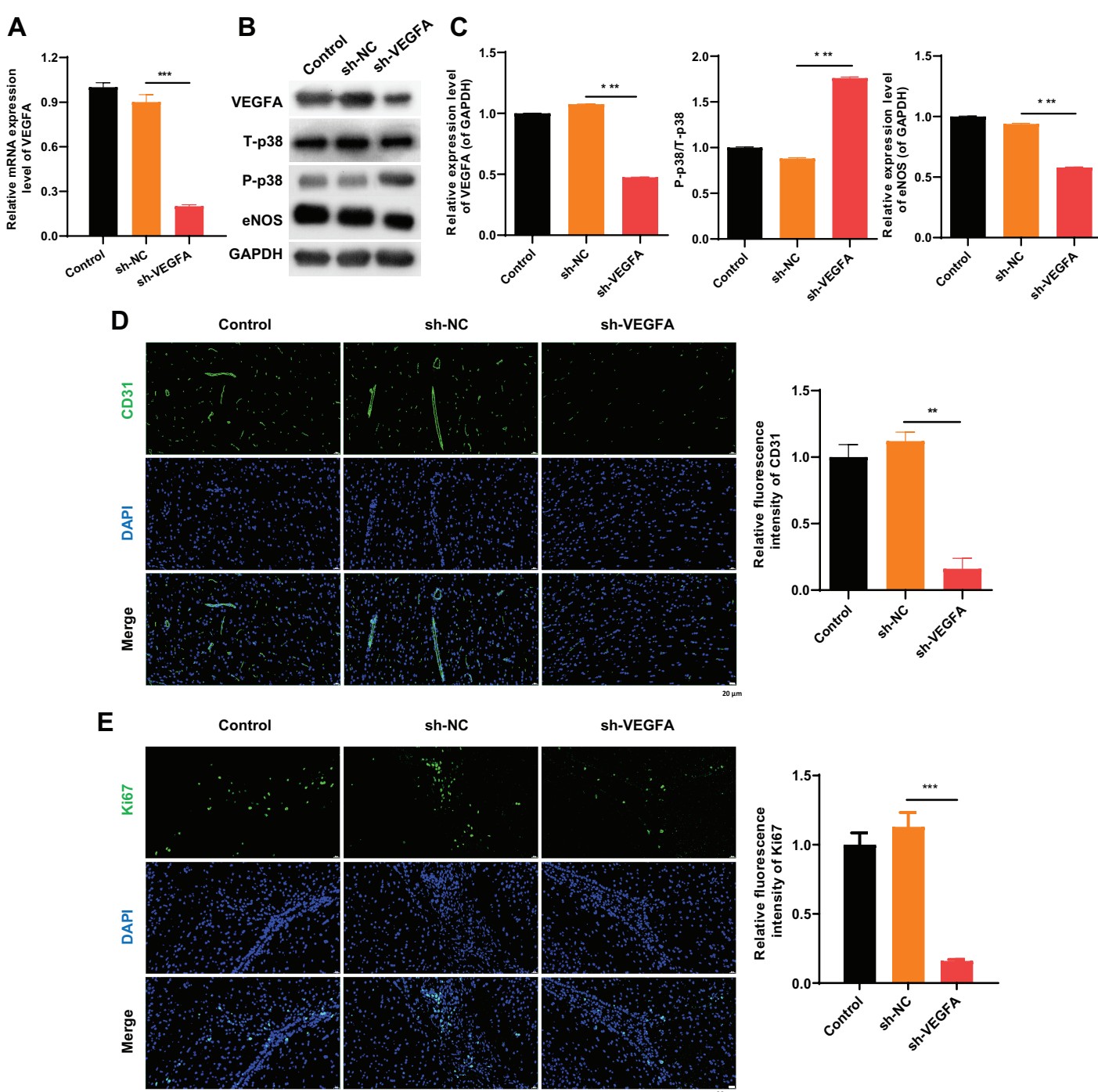

**Figure 4 VEGFA silencing led to an evident induction of p38 phosphorylation and eNOS downregulation in MCAO model rats mediated by β-asarone.** β-asarone-induced MCAO model rats were injected with adenovirus, including VEGFA shRNAs. (A) RT-qPCR was conducted to identify the change of VEGFA expression in the brain tissues. (B) Western blot was utilized to monitor the variation in expression of VEGFA, T-p38, P-p38, and eNOS. (C) Quantitative analysis of proteins was conducted using western blotting results. (D) CD31 and Ki67 were analyzed *via* IF double-staining assay of the brain tissues. Magnification: 400×, scale bar = 25 μm. (E) CD31 and Ki67 positive cells were quantitatively analyzed using IF results. $^{*}P < 0.05$, $^{**}P < 0.01$, $^{***}P < 0.001$. Control and MCAO model rats were administered β-asarone.

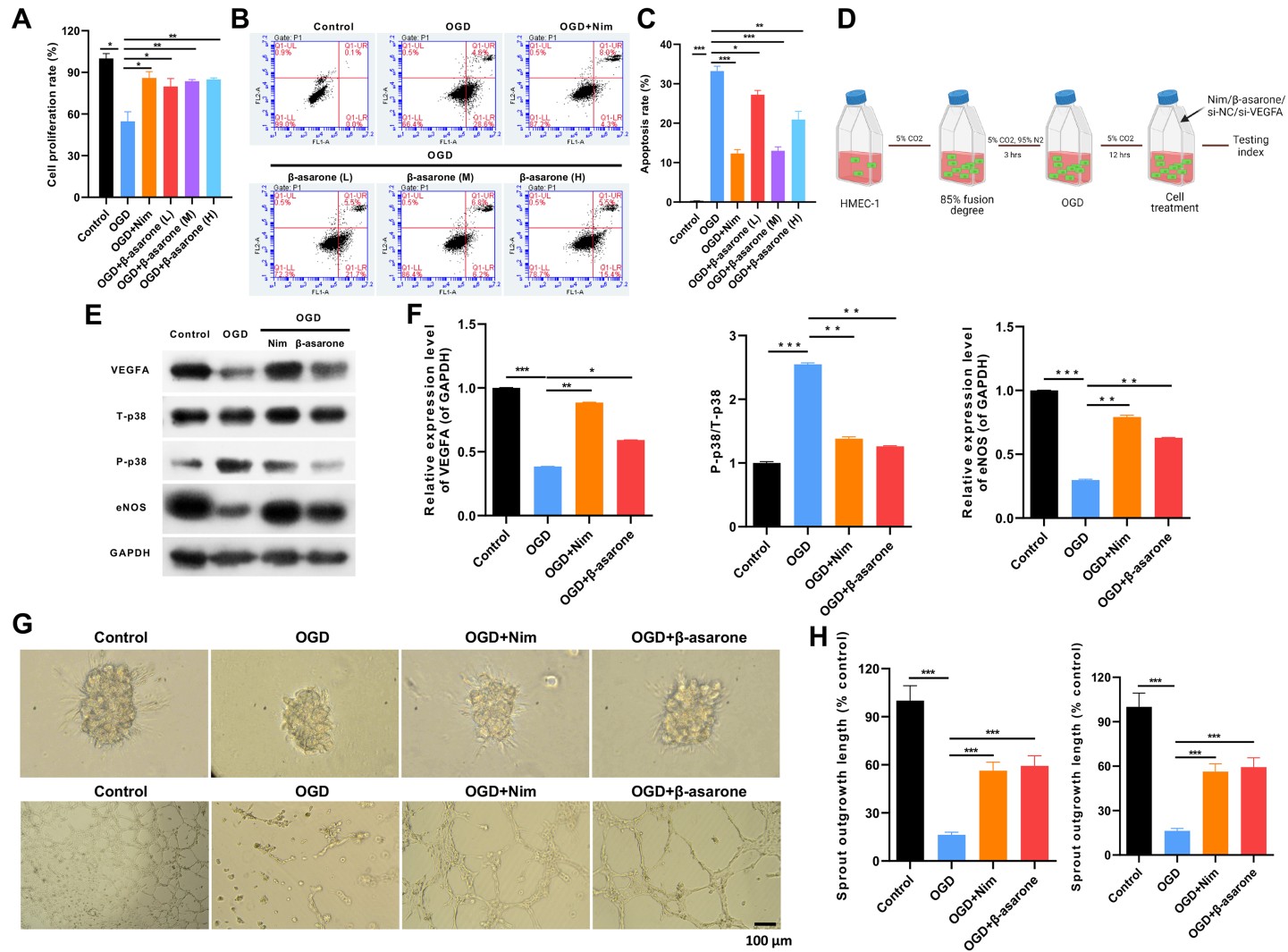

**Figure 5 β-asarone evidently accelerated cell viability and angiogenesis and suppressed apoptosis in HMEC-1 cells under oxygen-glucose deprivation (OGD).** OGD-treated HMEC-1 cells were administered nimodipine (10 μmol/L) or β-asarone (20, 30, and 45 μg/mL), respectively. (A) The change in cell viability of the processed HMEC-1 cells was examined using cell counting kit-8 (CCK-8). (B) Flow cytometry indicated a change in the apoptosis of HMEC-1 cells. (C) The apoptosis rate was statistically calculated. (D) The activity of HMEC-1 cells was analyzed. Image credit: the BioRender at https://app.biorender.com/. (E) Nimodipine (10 μmol/L) or β-asarone (30 μg/mL) were administered to treat HMEC-1 cells under OGD conditions, and VEGFA, T-p38, P-p38, and eNOS expressions were verified through western blot. (F) Each protein was quantitatively analyzed according to the gray value. (G) The changes in budding ability and tube-forming capacity in the treated HMEC-1 cells were analyzed using spheroid-based angiogenesis and tube-formation assays. (H) The total branching points and sprout outgrowth lengths were quantitatively analyzed. *$P < 0.05$, **$P < 0.01$, ***$P < 0.001$.

nimodipine or β-asarone (Figs. 5E and 5F). Meanwhile, it was confirmed that the total branching points (budding ability) were markedly lowered in the OGD group *versus* that in the control group, which could be attenuated by nimodipine or β-asarone in OGD-treated HMEC-1 cells (Figs. 5G and 5H). Additionally, the data from the tube formation assay indicated that the sprout outgrowth length (tube forming capacity) was also dramatically reduced in the OGD group *versus* that in the control group, which could also be significantly reversed by nimodipine or β-asarone in OGD-treated HMEC-1 cells

 

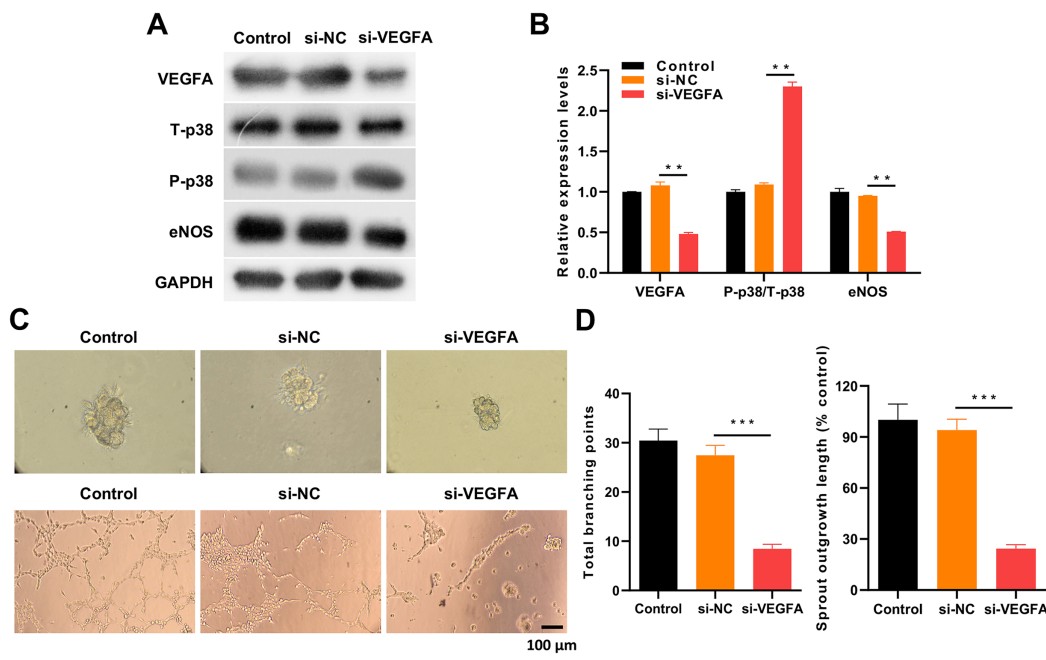

**Figure 6 VEGFA silencing led to a remarkable reverse of the impacts of β-asarone on VEGFA, p38, and eNOS pathways and angiogenesis in OGD-induced HMEC-1 cells.** OGD-induced HMEC-1 cells were administered 30 μg/mL β-asarone and transfected with si-VEGFA. (A) The changes in VEGFA, T-p38, P-p38, and eNOS expressions in cells were monitored through western blot. (B) The proteins were quantitatively analyzed *via* western blot. (C) Spheroid-based angiogenesis and tube formation assays were conducted to assess the influence of VEGFA silencing on the budding and tube-forming capacities. (D) The total branching points and sprout outgrowth length were quantitatively analyzed. $^{**}P < 0.01$, $^{***}P < 0.001$. Control and MCAO model rats were administered β-asarone.

(Figs. 5G and 5H). Thus, the study disclosed that β-asarone can also induce viability and angiogenesis and prevent apoptosis of HMEC-1 cells under OGD.

## VEGFA silencing evidently reversed the impacts of β-asarone on VEGFA, p38, and eNOS pathways and angiogenesis in OGD-induced HMEC-1 cells

Finally, the study also explored whether VEGFA could be involved in the induction of β-asarone on angiogenesis in OGD-induced HMEC-1 cells. β-asarone and si-VEGFA were utilized to treat OGD-induced HMEC-1 cells. Western blot analysis confirmed that VEGFA silencing significantly downregulated VEGFA and eNOS and induced p38 phosphorylation, which was mediated by β-asarone in OGD-treated HMEC-1 cells (Figs. 6A and 6B). Simultaneously, the data indicated that the budding ability induced by β-asarone could be significantly reduced by VEGFA silencing in OGD-treated HMEC-1 cells (Figs. 6C and 6D). VEGFA silencing also evidently weakened the tube-forming ability mediated by β-asarone in OGD-treated HMEC-1 cells (Figs. 6C and 6D). Consequently, the data confirms that the downregulation of VEGFA has a key role in the protective effect of β-asarone for HMEC-1 cells during OGD.

## DISCUSSION

IS is also called cerebral infarction (CI), which is ischemic necrosis or cerebromalacia caused by cerebral blood supply disorders, ischemia, and hypoxia (*Ma et al., 2023*; *Ruscu et al., 2023*; *Tang et al., 2020*). Presently, there exists no effective remedy to treat a stroke (*Barthels & Das, 2020*). Therefore, discovering effective therapy for IS is the central focus of current medical scholars and experts. To further investigate the possible drugs for effectively treating IS, an MCAO rat model was constructed following the methodology of previous research (*Zeng et al., 2023*). This study identified the expected cerebral ischemia injury in MCAO model rats. mNSS is the most commonly applied neurological function scale in IS animal studies to assess motor, sensory, reflex, and balance indicators (*Zhao et al., 2023a*). This study ascertained that the neurological function score of MCAO model rats markedly increased. Thus, MCAO model rats were constructed successfully. Meanwhile, the study also established OGD model cells using HMEC-1 cells by referring to relevant literature (*Zhan et al., 2023*). The data obtained reveals that OGD can significantly decrease cell viability and markedly increase apoptosis of HMEC-1 cells.

β-asarone has been reported to exhibit immunological, antibacterial, anticancer, myocardial protection, and other pharmacological effects (*Uebel et al., 2021*). It also demonstrates pharmacological effects on the central nervous system, digestive system, and cardiovascular system (*Hei et al., 2020*; *Saki et al., 2020*). When hypoxia occurs in organs, tissues, and cells, β-asarone can prevent oxidative phosphorylation and aggregate anaerobic metabolites, thus triggering a series of free radical reactions (*Hei et al., 2020*). Therefore, β-asarone can alleviate the hypoxia state of cells. Besides, research has confirmed that β-asarone can weaken autophagy during cerebral I/R injury by regulating JNK, p-JNK, Bcl-2, and Beclin 1 (*Liu et al., 2020a*). This study confirms that β-asarone can reduce infarct area and improve pathological injury in MCAO model rats. The study also investigated the effect of β-asarone on cell proliferation after cerebral infarction by dual immunofluorescence staining with endothelial cell-specific marker (CD31) and cell proliferation-specific marker (Ki67). It was proved that β-asarone can increase CD31 and Ki67 positive cells. Besides, β-asarone also induces cell viability and prevents apoptosis in OGD-treated HMEC-1 cells. Therefore, this study concludes that β-asarone holds a remarkable protective role on cerebral I/R injury. Studies have proved that nimodipine is clinically effective for treating IS, which can improve blood circulation during the recovery period of acute cerebrovascular disease (*Carlson et al., 2020*; *Ren et al., 2024*). This study utilized nimodipine as a positive control. The data showed that when the dose is appropriate, the protective effect of β-asarone is slightly weaker than that of nimodipine in MCAO model rats.

Previous research has confirmed that ischemia can cause degeneration and necrosis of massive neurons along with severe neurological defects (*Zhao et al., 2023b*). Angiogenesis of ischemic brain tissue and the addition of new collateral circulation can increase blood supply to improve ischemic brain function (*Hatakeyama, Ninomiya & Kanazawa, 2020*). Angiogenesis is a complex process, and the underlying function and mechanism of angiogenesis in IS are not fully understood. The data of this study further validate that β-

asarone can motivate the angiogenesis of HMEC-1 cells under OGD. Ki67 is a cell proliferation protein, which is presently the most reliable indicator reflecting cell proliferation activity and rate (*Nadeem et al., 2023*). CD31 is a prognostic angiogenic marker involved in cell angiogenesis (*Caligiuri, 2020*). Ki67 and CD31 reflect cell proliferation and angiogenesis. Interestingly, this study discovered that β-asarone can increase CD31 and Ki-67 positive cells in MCAO model rats. Therefore, this confirms that β-asarone can induce angiogenesis in MCAO model rats.

VEGF, the strongest angiogenic stimulator, mainly contains VEGFA (vascular permeability factor), VEGFB, VEGFC, VEGFD, and placental growth factor (PIGF) in mammalian cells (*Liang & Zhao, 2021*). VEGFA is a major member of the VEGF family involved in angiogenesis, and it is a target of anti-angiogenesis therapy (*Zhang et al., 2023*). VEGFA has been reported to participate in vasodilation, proliferation, permeability, migration, and survival in cells (*Wiszniak & Schwarz, 2021*). Existing evidence suggests that VEGFA has the strongest angiogenic activity (*Pérez-Gutiérrez & Ferrara, 2023*), thus acting as one of the main factors in this study. VEGFA can be tested in plentiful tissues *in vivo* under physiological conditions (*Braile et al., 2020*; *Bujaldon et al., 2019*). Besides, VEGFA expression can be regulated by multiple factors, with hypoxia being the most crucial regulatory factor (*Manukjan et al., 2023*). Several studies also demonstrated that VEGFA can participate in multiple pathophysiological processes, such as atherosclerosis, collateral circulation formation, increased vascular permeability, brain edema, neuroprotection, and nerve regeneration in ischemic cerebrovascular disease (*Manukjan et al., 2023*; *Neill et al., 2020*; *Parab et al., 2023*). It has been affirmed that cardamonin can alleviate cerebral ischemia/reperfusion (I/R) injury by inducing the VEGFA pathway (*Ni et al., 2022*). Long non-coding RNA (LncRNA) MEG8 also reduced cerebral ischemia and upregulated VEGFA in MCAO rats (*Sui et al., 2021*). In general, VEGFA exerts a protective effect on cerebral I/R injury. This study proves that β-asarone can upregulate VEGFA in MCAO model rats and OGD-induced HMEC-1 cells. Additionally, it was demonstrated that VEGFA silencing could accelerate cerebral infarction and pathological injury, which was weakened by β-asarone in MCAO model rats. β-asarone facilitates the angiogenesis of OGD-induced HMEC-1 cells by elevating VEGFA expression. Thus, this study substantiates that β-asarone can mitigate IS by upregulating VEGFA to induce angiogenesis.

Mitogen-activated protein kinase (p38 MAPK) is a vital signal transduction molecule, has been associated with multiple physiological processes, including cell growth, proliferation, differentiation, death, and so on (*Falcicchia et al., 2020*; *Martínez-Limón et al., 2020*). p38 also exhibits pleiotropy and high efficiency in inflammatory response, which makes it a new target for the study of inflammation-related diseases (*Awasthi, Raju & Rahman, 2021*). Inflammation is also connected with angiogenesis (*Jeong, Ojha & Lee, 2021*). p38, same as VEGFA, is also key in regulating angiogenesis (*Zhou et al., 2023*). Research also showed that VEGFA can activate the p38 pathway and induce the phosphorylation of p38 to regulate cell angiogenesis (*Cheng et al., 2023*). MiR-497-5p could aggravate vascular endothelial cell dysfunction by targeting VEGFA/p38/MAPK pathway in atherosclerosis (*Lu et al., 2024*). These data suggest that VEGFA can regulate

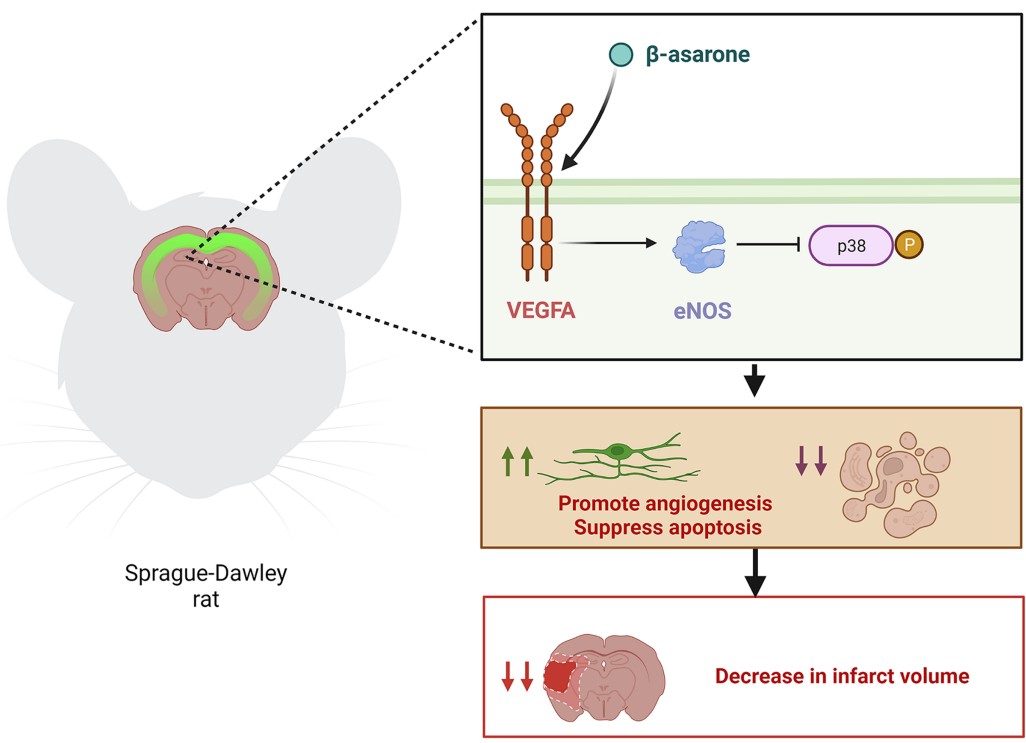

**Figure 7 Diagrammatic representation of β-asarone, which can alleviate brain injury in MCAO model rats.** β-asarone can decrease the infarct volume of MCAO model rats by utilizing VEGFA to promote angiogenesis and suppress apoptosis. Image credit: the BioRender at https://app.biorender.com/.

p38 and affect angiogenesis. Moreover, VEGFA can promote the expression and activity of eNOS by activating VEGF receptor-2 (KDR), thereby increasing the amount of nitric oxide produced by endothelial cells (*Kwak et al., 2006*). This mechanism is essential for VEGFA-induced angiogenesis and endothelial function. Besides, VEGFA can affect eNOS function through other pathways, such as by changing the phosphorylation status of eNOS to further enhance its ability to produce nitric oxide (*Bouloumié, Schini-Kerth & Busse, 1999*). eNOS is an enzyme that catalyzes the production of nitric oxide (NO) (*Guo et al., 2023*). NO, as a potent vasodilator, can improve blood flow, reduce vascular inflammation, and inhibit apoptosis (*Chen et al., 2023*). Apoptosis is one of the key factors leading to neuronal death in patients with IS (*Liu et al., 2023*). Thus, activating the VEGFA/eNOS pathway can weaken apoptosis and improve neurological recovery after cerebral ischemia (*Liu et al., 2020b*). Besides, the experiment also proved that VEGF is affected by the PI3K/ AKT pathway (*Lu et al., 2020*). eNOS is the first and only molecule affected by AKT to promote angiogenesis (*Yu et al., 2020b*). This study proves that β-asarone can suppress p38 phosphorylation and upregulate eNOS in MCAO model rats and OGD-induced HMEC-1 cells by upregulating VEGFA.

The results of this study may apply to other species (*e.g.*, mice, guinea pigs, rabbits, *etc.*) or experimental conditions. However, there are limitations to the current study. For instance, the sample size in each group was not large enough. This study only validated

VEGFA. The VEGF family includes other members, such as VEGFB, VEGFC, and VEGFD, each of which also has specific biological functions and mechanisms of action. Therefore, future studies can explore the specific roles of VEGF family members in IS to facilitate designing more effective therapeutic approaches. The results of this study emphasize the complex molecular mechanisms involved in the neuroprotective effects of β-synuclein, such as VEGFA, eNOS, and p-p38. However, the specific interactions and downstream effects of these proteins are not clear. Future mechanistic studies may provide new insights into potential therapeutic strategies for IS. This underscores an important direction for our future research. Moreover, this study preliminarily proves that β-asarone can serve as a potential treatment for IS, but further preclinical and clinical trials are necessary to validate the efficacy and safety of β-asarone in patients with IS.

## CONCLUSION

The present study demonstrates that β-asarone evidently exhibits a therapeutic effect on IS, which can induce viability and angiogenesis and suppress apoptosis. Besides, the study proves that VEGFA significantly contributes to the protective effect of β-asarone on IS. Therefore, the study concludes that β-asarone can play a central role in treating IS, with VEGFA being a potential therapeutic target for IS (Fig. 7).

### Funding

This study was supported by grants from the National Key R&D Program of China, Ministry of Science and Technology of China (2018YFC2002504). The funders had no role in study design, data collection and analysis, decision to publish, or preparation of the manuscript.

### Grant Disclosures

The following grant information was disclosed by the authors:
National Key R&D Program of China.
Ministry of Science and Technology of China: 2018YFC2002504.

### Competing Interests

The authors declare that they have no competing interests.

### Author Contributions

- Dazhong Sun conceived and designed the experiments, performed the experiments, analyzed the data, prepared figures and/or tables, authored or reviewed drafts of the article, and approved the final draft.
- Lulu Wu performed the experiments, prepared figures and/or tables, and approved the final draft.
- Siyuan Lan analyzed the data, prepared figures and/or tables, and approved the final draft.

- Xiangfeng Chi analyzed the data, prepared figures and/or tables, authored or reviewed drafts of the article, and approved the final draft.
- Zhibing Wu conceived and designed the experiments, authored or reviewed drafts of the article, and approved the final draft.

### Animal Ethics

The following information was supplied relating to ethical approvals (*i.e.*, approving body and any reference numbers):

All animal experiments were approved by the Ethics Committee of the first affiliated hospital of Guangzhou University of Chinese Medicine.

### Data Availability

The data is available at figshare: Wu, Zhibing (2024). Raw data. figshare. Journal contribution. https://doi.org/10.6084/m9.figshare.24947667.v2.

### Supplemental Information

Supplemental information for this article can be found online at http://dx.doi.org/10.7717/peerj.17534#supplemental-information.

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
