# Peer review of "β-asarone induces viability and angiogenesis and suppresses apoptosis of human vascular endothelial cells after ischemic stroke by upregulating vascular endothelial growth factor A"

_PeerJ, doi:10.7717/peerj.17534_

## Round 0.1 · original submission · Major Revisions

Authors should revise according to the suggestions of reviewers. The modifications should be marked. A point-to-point response letter is needed.

**Language Note:** The review process has identified that the English language must be improved. PeerJ can provide language editing services - please contact us at [email protected] for pricing (be sure to provide your manuscript number and title). Alternatively, you should make your own arrangements to improve the language quality and provide details in your response letter. – PeerJ Staff

Reviewer 1 ·

Basic reporting

The paper explored β-asarone induces viability and angiogenesis and suppresses apoptosis following ischemic stroke by upregulating vascular endothelial growth factor A. Authors should attempt to describe potential mechanisms for the interaction of VEGFA/eNOS pathway and apoptosis more precisely.

Experimental design

In materials and methods section, procedure and administration of drugs etc. in need to be described in detail, add references if necessary.How about quantitative analysis of CD31 and Ki67 positive cells?
VEGF mainly contains VEGFA , VEGFB, VEGFC, VEGFD and PIGF, and the corresponding receptors are also numerous and complex. Why was only VEGFA studied in this project, were other factors and receptors considered, the authors should clarify this issue.Also, the source or target of action of the VEGFA pathway needs to be stated, is it directed at neurons? Glial cells? or vascular endothelial cells?

Validity of the findings

This study is potentially interesting,and the data is accurate.
VEGF mainly contains VEGFA , VEGFB, VEGFC, VEGFD and PIGF, and the corresponding receptors are also numerous and complex. Why was only VEGFA studied in this project, were other factors and receptors considered, the authors should clarify this issue.Also, the source or target of action of the VEGFA pathway needs to be stated, is it directed at neurons? Glial cells? or vascular endothelial cells?

Additional comments

None.

Reviewer 2 ·

Basic reporting

This study explores the potential role of β-asarone in the treatment of ischemic stroke, providing empirical evidence through animal models and in vitro experiments, thus offering strong support for further investigation of β-asarone as a therapeutic agent for ischemic stroke. This paper is informative, but there are still some issues that need to be addressed. Here are some comments:
1)Acronyms appearing for the first time in Abstract 1, Body 2, and Figure/Figure note/Table 3 need to be supplemented with the full name. Please check all the acronyms.
2)The research topic and methods described in the abstract are clear and concise. However, it is suggested to briefly introduce the background of ischemic stroke (IS) at the beginning of the abstract to provide readers with better context.
3)The language in the manuscript needs to be polished to enhance clarity and precision of expression, ensuring smoother comprehension for the readers.
4)In the second paragraph of the introduction, what is the relationship between α-asarone and β-asarone.
5)It should be stated in the Materials and Methods section that nimodipine is used as a positive control drug.

Experimental design

The detailed nature of the experimental methods is crucial for other researchers to replicate and validate the results. But some methods was insufficient:
1)Please provide the sequences of si-NC and si-VEGFA.
2)In the western blot, please provide dilution concentrations and item numbers of primary and secondary antibodies.
3)The transfection steps should be described in more detail.

Does the selection of drug dosage and the method of adenovirus injection have literature support?

To ensure consistency in data analysis, it is necessary to define the criteria for assessing infarct size and the method for determining the infarct area.

Validity of the findings

-The author mentioned in the description of the TTC staining method that the brain tissue was then cut into slices with a thickness of 2 mm and subsequently stained. However, the TTC staining images presented in Figure 1B and Figure 3B do not appear to be slices of brain tissue.

-The font in the figures should be consistent, label text should not be obscured, and the text arrangement should be neat. Please check all the images.

-The legend in Figure 1 needs to define the meaning of the “*”.

-Figure 2D, 4D, and 5G should have a scale bar.

-The staining results for Ki67 are quite confusing, as positive expression is hardly visible, which raises doubts about the reliability of the results.

-In the conclusion section, the authors elicited VEGFA activators. However, the effect of VEGFA activators on IS was not investigated in this study. Authors should revise the conclusion section to relate it to the results and limit the conclusions to those supported by the results.

Additional comments

No comment.

Reviewer 3 ·

Basic reporting

This study aimed to investigate the potential of β-asarone in improving ischemic stroke (IS) injury by inducing angiogenesis. Using a rat middle cerebral artery occlusion (MCAO) model and oxygen-glucose deprivation (OGD) model cells, we evaluated the effects of β-asarone on cerebral infarction, pathological damage, cell viability, apoptosis, and angiogenesis. Our results demonstrated that β-asarone, similar to nimodipine, mitigated cerebral infarction and pathological damage in MCAO rats. Additionally, β-asarone upregulated vascular endothelial growth factor A (VEGFA) and endothelial nitric oxide synthase (eNOS) expression, as well as induced p38 phosphorylation in MCAO rats. In vitro experiments showed that β-asarone enhanced cell viability, reduced apoptosis, and promoted angiogenesis by upregulating VEGFA in OGD-treated human microvascular endothelial cells (HMEC-1). These findings suggest that β-asarone protects against IS injury by upregulating VEGFA expression, indicating its potential as a therapeutic agent for ischemic stroke. However, the manuscript is subject to the necessary revisions before being recommended for acceptance, as follows:
1.The manuscript will benefit from further proofreading of the English writing, giving the reader a clearer understanding of the author's intentions.
2.The citations in the references mainly focus on 2019-2022, which is generally appropriate. However, the authors should appropriately search the latest research progress in 2023 or even 2024 for updates.
3.The Introduction describes the multiple therapeutic functions and benefits of Acorus TatarinoWii Schott, particularly focusing on the active component β-asarone. β-asarone has been reported to have various pharmacological activities, including antioxidant, anti-inflammatory, and neuroprotective effects. Clinical use of β-asarone is indicated for conditions such as fever, delirium, dementia, and neurological disorders like Alzheimer's disease and epilepsy. Previous studies have shown that β-asarone can cross the blood-brain barrier, prevent neuronal apoptosis, enhance cognitive function, and alleviate cerebral infarction in animal models. The study aims to investigate the potential role of β-asarone in ameliorating ischemic stroke (IS) injury, with a focus on its effects on angiogenesis. Given the importance of angiogenesis in post-stroke recovery, the study seeks to elucidate whether β-asarone can induce angiogenesis in IS. Therefore, the authors should clarify the role of angiogenesis in IS.
3. Abbreviated words in the article need to be written in full when they first appear, such as TTC, H&E, MCAO, and OGD.
4. The information on some reagents in the Material and Method is not detailed, and some experimental methods are too simple and need to be refined.
5. The study in the Results suggests that the protective effects of β-asarone on brain injury in MCAO rats may be mediated through the regulation of VEGFA, p38 MAPK, and eNOS pathways. These pathways are crucial for angiogenesis, neuroprotection, and cellular proliferation, indicating the potential mechanisms underlying the beneficial effects of β-asarone in ischemic stroke. The author needs to clarify why VEGFA, p38 MAPK, and eNOS pathways were chosen for the study.
6. The findings indicate that VEGFA silencing reverses the regulatory effects of β-asarone on p38 phosphorylation, eNOS expression, and angiogenic/proliferative markers in MCAO model rats. These results highlight the intricate molecular mechanisms involved in the neuroprotective actions of β-asarone, with VEGFA emerging as a key mediator in modulating signaling pathways essential for ischemic stroke pathophysiology. Further research into the specific interactions and downstream effects of these proteins may offer novel insights into potential therapeutic strategies for ischemic stroke management.
7. While the results show promise for β-asarone as a potential therapy for IS, further studies, including preclinical and clinical trials, are necessary to validate the efficacy and safety of β-asarone in human patients with IS.
8. The discussion needs further refinement, highlighting the focus of the research, and conducting in-depth discussions on innovative points in conjunction with existing reports.

Experimental design

no comment

Validity of the findings

no comment

---

## Round 0.2 · Major Revisions

Authors should revise according to the suggestions of reviewers. The modifications should be marked. A point to point response letter is needed.

Reviewer 1 ·

Basic reporting

no comment.

Experimental design

The staining results for Ki67 and CD31 are quite confusing, references need to be provided, while poor quality images need to be replaced.

Validity of the findings

The staining results for Ki67 and CD31 are quite confusing, references need to be provided, while poor quality images need to be replaced.

Additional comments

The author needs to clarify why VEGFA, p38 MAPK, and eNOS pathways were chosen for the study.How the VEGFA, p38 MAPK, and eNOS signalling pathway is linked and its lack of evidence.

Reviewer 2 ·

Basic reporting

The authors have addressed my concerns thoroughly in their revisions, and I recommend acceptance for publication.

Experimental design

no comment

Validity of the findings

no comment

Additional comments

no comment

Reviewer 3 ·

Basic reporting

The author has made a relatively comprehensive revision and response to the comments. I think they have solved my doubts well, and the manuscript has been greatly improved.

Experimental design

no comment

Validity of the findings

no comment

---

## Round 0.3 · accepted · Accept

I think this manuscript was well organized and it could be accepted.

Reviewer 1 ·

Basic reporting

No comment.

Experimental design

No comment.

Validity of the findings

No comment.

Additional comments

Authors have made revisions accordingly. The original manuscript is eligible for publication.